# Implementing discrete-time quantum walks on multi-dimensional graphs in cavity quantum electrodynamics

Qi-Ping Su[1], Chen-Hui Peng[1], Li Yu[1], Wei Feng[1], Guo-Qiang Zhang[1] and Chui-Ping Yang[1*]

**1** School of Physics, Hangzhou Normal University, Hangzhou, Zhejiang 311121, China

* yangcp@hznu.edu.cn

November 3, 2024

## Abstract

One of the final goals in quantum information science is to achieve large-scale quantum computing. Discrete-time quantum walks (DTQWs) can be used to realize universal quantum computing. Based on cavity quantum electrodynamics (CQED), previous research focuses on implementing DTQWs on one-dimensional simple graphs. To implement large-scale quantum computing, it is necessary and urgent to realize DTQWs on multi-dimensional graphs with different structures. We propose a general protocol for realizing DTQWs on multi-dimensional graphs with CQED, where each graph node can have a different number of connected neighbor nodes. As an application, we numerically simulate a Grover walk search algorithm in a cubic graph using superconducting devices. With decoherence considered, our simulation results fit well with the theoretical results. The protocol is universal and can be extended to accomplish the same task in a wide range of CQED systems, which consist of natural or artificial atoms and optical or microwave cavities. This work paves an avenue to realize DTQWs on multi-dimensional graphs, which could have broad applications in large-scale quantum computing and quantum simulation.

# 1 Introduction

Cavity quantum electrodynamics (CQED) [1] studies the interaction between natural or artificial atoms and quantized electric and magnatic fields stored in cavities, which is essential for quantum information processing. Based on CQED, a large number of experiments have been conducted in various quantum systems, such as the superconducting (SC) system [2] and the trapped ion system [3], which result in many interesting applications [4–7].

As the extension of classical random walks, discrete-time quantum walks (DTQWs) have important applications in universal quantum computation [8–13], quantum algorithm [14–18], quantum simulation [19–23], quantum state engineering [24, 25], etc. In a standard DTQW, a walker moves in a graph consisting of nodes and edges, depending on the state of the coin. Each edge acts as a bridge connecting adjacent nodes. It is commonly recognized that the implementation of DTQWs in solid-state quantum systems is not easy. Unlike photons [26] or atoms [27], solid-state qubits (such as superconducting qubits) can not move because their positions are fixed in space. Thus, it is difficult to encode states of the coin and positions of the walker by solid-state qubits.

Up to today, there exist only a few schemes for realizing DTQWs on one-dimensional (1D) graphs based on CQED [28–33]. Let us here give a brief introduction to Refs. [28–33]. In [28–30], the phase space of superposition Fock states in a cavity was used to encode the walker's position space and a qubit (coupled with the cavity) was used as the coin. Due to the use of non-orthogonal states and the limitation of the phase space, the generality and the scalability of [28–30] pose inevitable problems. In [31], a 1D DTQW scheme was proposed, where a pair of superconducting qubits were used as a node and the nearest-neighbor qubits were coupled via tunable couplers. The walker moved among nodes in a 1D line and the coin was encoded using excited energy levels of the qubit in each node. In [32], qutrits (i.e., three level quantum systems) were used to realize a 1D DTQW, where a 2D coin was encoded by two higher energy levels of the qutrits. In [33], quantum walks of a single particle in a 1D periodically kicked lattice were investigated.

Note that the previous works on 1D DTQWs [28–33] can only be used to realize quantum computing with a small size. As is well known, one of final goals of quantum computing is the implementation of large-scale quantum computation. For quantum computation based on DTQW, large-scale quantum computation would require implementing DTQWs on a multidimensional graph. However, after a careful review of the literature, we find that how to realize a DTQW on a multiple-dimensional (MD) or complex graph based on CQED has not been reported to date. We stress that it is much more difficult to realize a MD DTQW than a 1D DTQW, because a MD DTQW requires much more nodes to encode the positions of the walker and a higher dimensional space to encode the coin. Thus, implementing a MD DTQW is challenging in most physical systems, especially in solid-state systems because solid-state qubits can not hop in space. On the other hand, the operations in qudits (i.e., quantum systems with more than two energy levels) and their applications in quantum algorithms have attracted considerable interest in recent years, with experimental realizations achieved in various systems [34–40]. Compared to a two-dimensional qubit, a qudit offers a higher-dimensional space for encoding more information.

In this paper, we propose a general protocol for implementing DTQWs on MD graphs (i.e., MD DTQWs) based on CQED using qudits, where each graph node can have a different number of connected neighbor nodes. In this proposal, graph nodes are constructed by qudits while graph edges are constructed by cavities. Positions of the walker are encoded by sites of qudits in a graph, while states of the coin are encoded by excited states of the qudits. Specifically, an $N$-dimensional coin should be used when the walker is located at a site connecting to $N$ neighbouring sites, i.e., the walker at this site can move in $N$ directions. In this case, with

the ground energy level included, a qudit with $(N + 1)$ energy levels is required at this site. Note that the $N$ excited energy levels of this qudit are used to encode the coin at this site. Cavities or resonantors are used as graph edges to control the coupling or decoupling between the neighbor qudits.

As an application of this protocol, we numerically simulate a Grover walk search algorithm in a cubic graph using superconducting qudits and cavities. With decoherences considered in the simulations, we study the effect of the systematical parameters on the optimal probability, i.e., the probability for successfully finding the target state in the final state after a two-step DTQW. The results indicate that this protocol is feasible within the current circuit QED technology.

## 2 General Protocol for Implementing DTQWs on MD graphs

For a standard DTQW on a MD graph, a walker moves among graph nodes with respect to the state of a coin. The joint evolution of the walker and the coin in each step is characterized by a unitary operator $U = S \cdot C$, where $C$ is the operator describing the operation on the coin while $S$ is the operator for the walker. Note that the coin can have different dimensions when the walker is located at different nodes. In each step of a standard DTQW, the coin is first tossed by the operator

$$C = \sum_{k=1}^{n} C_k \otimes |k\rangle_w \langle k|, \tag{1}$$

where $C_k$ is the $N_k$-dimensional unitary operator at node $k$, given by

$$C_k = \left( \sum_{i_k=1}^{N_k} \sum_{j_k=1}^{N_k} \alpha_{i_k j_k} |i_k\rangle_c \langle j_k| \right). \tag{2}$$

Here and above, $n$ is the number of graph nodes, $N_k$ is the dimension of the coin at node $k$ (i.e., the number of directions in which the walker at node $k$ can move), $|i_k\rangle_c$ represents the $i_k{}^{th}$ basic state of the coin at node $k$, and the subscript "$c$" ("$w$") stands for the coin (walker). In addition, the walker is shifted by the operator

$$S = \sum_{k=1}^{n} S_k, \tag{3}$$

where $S_k$ is the shift operator for the walker at node $k$, which is associated with the coin state, and given by

$$S_k = \sum_{i_k=1}^{N_k} |i_k\rangle_c \langle i_k| \otimes |i_k\rangle_w \langle k|. \tag{4}$$

Here $|i_k\rangle_w$ represents the position of the $i_k{}^{th}$ neighbor node of node $k$, as shown in Fig. 1. Note that the position of the walker is represented by the node occupied by the walker.

Based on cavity QED, we now propose a general protocol for implementing DTQWs on a MD graph, described by the above. In this proposal, graphs for DTQW are constructed by qudits and cavities. Namely, qudits serve as graph nodes while cavities play a role as the graph edges to connect the nodes.

We first introduce how to implement $C_k$ and $S_k$ on a subgraph composed by node $k$ and its $N_k$ neighbor nodes ($N_k$ can be an *arbitrary* positive integer), with the walker initially at the position of node $k$ (see Fig. 1). Note that in each step of a standard DTQW, a walker shifts from its located node to its neighboring nodes only. In addition, MD graphs can be decomposed

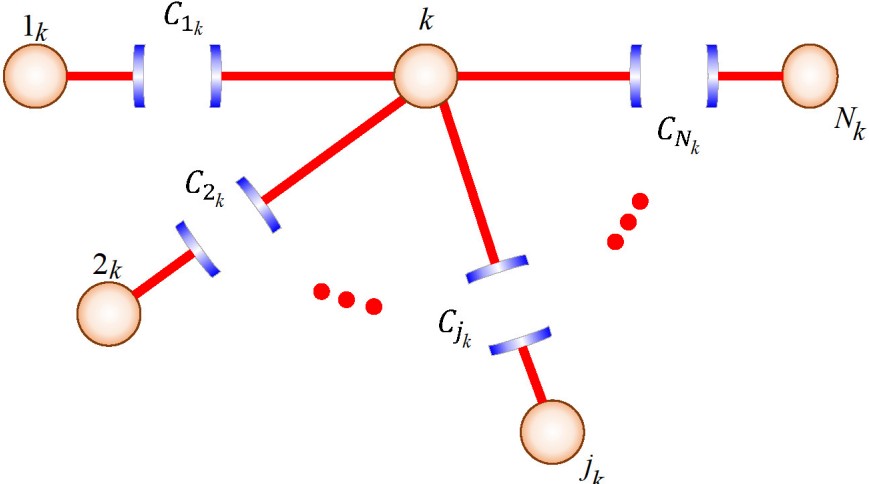

Figure 1: Diagram of a qudit at node $k$ and its $N_k$ neighbor qudits $(1_k, 2_k, ..., N_k)$. The qudit at node $k$ and its neighbor qudit $j_k$ are coupled via cavity $C_{j_k}$ $(j_k = 1_k, 2_k, ..., N_k)$. Each qudit has an identical energy level structure, while frequencies of the $N_k$ cavities are different from each other.

as nodes and their edges connecting the neighbors. Therefore, if the subgraph (shown in Fig. 1) can be constructed in cavity QED and the one-step DTQW operator $U_k = S_k \cdot C_k$ can be successfully implemented in this subgraph, a complete graph with different structure can be constructed in the same manner and a multi-step DTQW can be realized by repeating the one-step DTQW in each subgraph (consisting of a node and its neighbor nodes).

The construction of the subgraph is shown in Fig. 1, where qudit $k$ (at node $k$) interacts with its neighboring qudits $1_k, 2_k, ..., N_k$ via cavities $C_{1_k}, C_{2_k}, ..., C_{N_k}$, respectively. Assume that the walker is initially located at node $k$. In this case, one needs to adopt $N_k + 1$ energy levels of qudit $k$, which are labeled as $|0\rangle_k, |1\rangle_k, ...,$ and $|N_k\rangle_k$. The $N_k$ excited energy levels of qudit $k$ are used to encode an $N_k$-dimensional coin, which is required when the walker is at position $k$. A coin with an $N_k$ dimensional size is needed to have the walker move in $N_k$ different directions. For simplicity, we assume that the $N_k$ neighboring qudits $1_k, 2_k, ..., N_k$ of qudit $k$ have the same energy level structure as qudit $k$ (Fig. 1). In general, one can set the number of energy levels of each qudit to be equal to the number of its neighbors plus 1, or just set the number of energy levels of all qudits to be equal to the maximal number of neighbors of nodes plus 1.

In Fig. 1, the position of the walker is encoded by the location of a qudit, which is not in the ground state. For instance, if qudit $k$ or $j_k$ $(j_k = 1_k, 2_k, ..., N_k)$ is not in the ground state, then the walker is at position $k$ or $j_k$. Let us suppose that the walker is initially at position $k$ (the location of qudit $k$ in Fig. 1). In this case, all qudits are initially in the ground state except qudit $k$. In addition, assume that all cavities are initially in the ground state (i.e., the vacuum state).

We now introduce how to implement a one-step DTQW on the subgraph shown in Fig. 1, which is described by the operator $U_k = S_k \cdot C_k$. Here, $C_k$ characterizes the tossing of the coin at walker's position $k$ while $S_k$ describes the moving of the walker into $N_k$ neighbor positions $1_k, 2_k, ..., N_k$ according to the coin states $|1\rangle_c, |2\rangle_c, ..., |N_k\rangle_c$. For the expressions of $C_k$ and $S_k$, please see Eqs. (2) and (4). This one-step DTQW on the subgraph can be implemented by the following two processes.

**Process I - Tossing the coin with classical pulses.** First, adjust the level spacings of each qudit to have all qudits decouple from the cavities. Then, apply classical pulses to each qudit

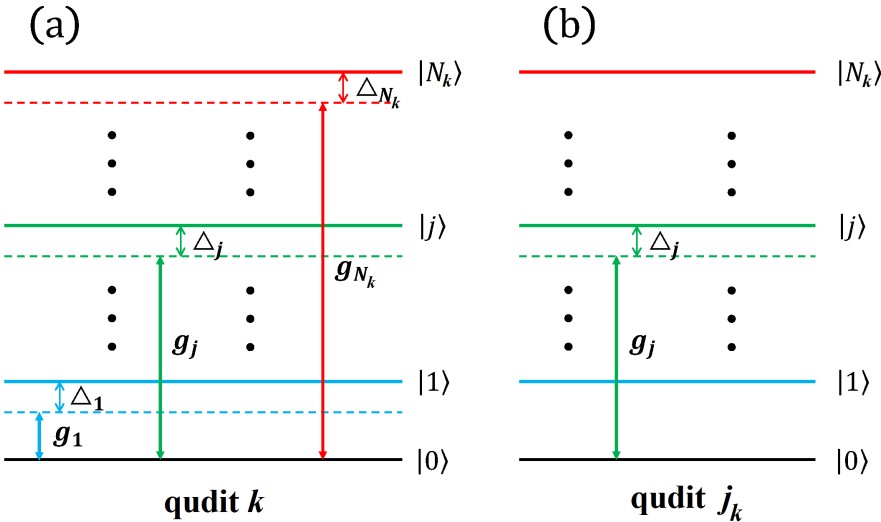

Figure 2: Illustration of cavity $C_{j_k}$ ($j_k = 1_k, 2_k, ..., N_k$) being coupled to the $|0\rangle \leftrightarrow |j\rangle$ transition of qudit $k$ ($j_k$) with coupling strength $g_j$ ($\mu_j$) and detuning $\Delta_j$. (a) The couplings of the $N_k$ cavities ($C_{1_k}, C_{2_k}, ..., C_{N_k}$) with qudit $k$. (b) The couplings of cavity $C_{j_k}$ with qudit $j_k$.

to realize the coin operator $C_k$. More explanations are given below.

According to the CS (Chandler and Stewart) decomposition [41], a $SU(N_k)$ operation on the $N_k$-dimensional coin, described by the coin operator $C_k$, can be decomposed into $SU(2)$ and $SU(3)$ operations. Moreover, an arbitrary $SU(3)$ operation can be decomposed into $SU(2)$ operations and $U(1)$ operations. Here, each $U(1)$ operation only adds a phase $e^{i\phi_0}$ to one basis state of the coin. In this way, a $SU(N_k)$ operation on the $N_k$-dimensional coin can be implemented through $SU(2)$ and $U(1)$ operations, which can be realized by applying classical pulses. An arbitrary $SU(2)$ operation can be expressed as a 2D matrix of form

$$\begin{pmatrix} \cos\theta & -ie^{-i\phi}\sin\theta \\ -ie^{i\phi}\sin\theta & \cos\theta \end{pmatrix}. \tag{5}$$

A $SU(2)$ unitary operation, acting on the two excited states of $|j\rangle$ and $|l\rangle$ of a qudit, can be realized by applying a classical pulse resonant with the $|j\rangle \leftrightarrow |l\rangle$ transition of the qudit. The qudit-pulse resonant interaction results in the following state rotations [42]

$$|j\rangle \rightarrow \cos(\Omega t)|j\rangle - ie^{-i\phi}\sin(\Omega t)|l\rangle,$$
$$|l\rangle \rightarrow -ie^{i\phi}\sin(\Omega t)|j\rangle + \cos(\Omega t)|l\rangle, \tag{6}$$

where $\Omega$ is the pulse Rabi frequency and $\phi$ is the initial phase of the pulse. By setting the operational time $t = \theta/\Omega$, one can see that the state transformation (6) can be described by the $SU(2)$ operation given in Eq. (5). In this sense, the $SU(2)$ operation (5) is implemented. In general, the $U(1)$ operation on an energy level $|h\rangle$ is rarely used, which can be realized by applying a classical pulse off-resonant with the $|0\rangle \leftrightarrow |h\rangle$ transition of the qudit.

After the above process, the unitary operator $C_k$ of the coin (i.e., the $SU(N_k)$ operator) is realized, which prepares the qudit $k$ and its $N_k$ neighbor qudits in the following state

$$\left( \sum_{j=1}^{N_k} \alpha_j |j\rangle_k \right) \prod_{m_k=1_k}^{N_k} |0\rangle_{m_k}, \tag{7}$$

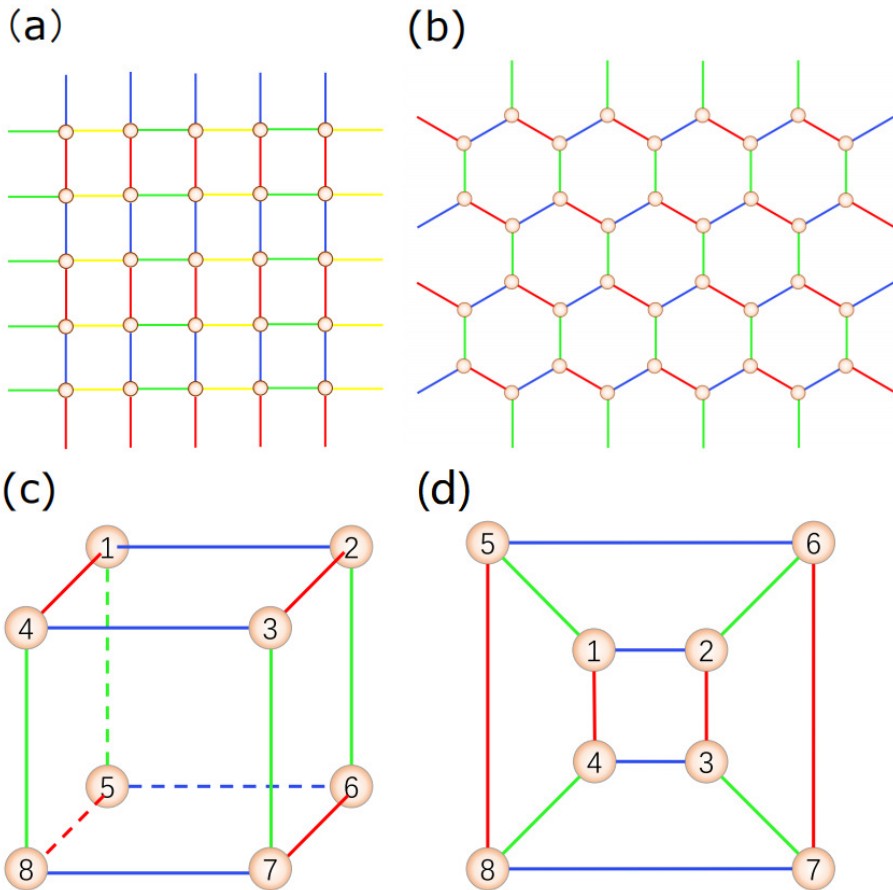

Figure 3: Three types of 2D graphs. Each dot represents a qudit while each line represents a cavity. The lines with the same color represent the cavities with the same frequency. Every qudit couples to its adjacent cavities with different frequencies.

where the subscript $k$ represents qudit $k$ while the subscript $m_k$ represents its neighbor qudit $m_k$ ($m_k = 1_k, 2_k, ..., N_k$).

**Process II - Shifting the walker from position $k$ to its neighbor positions according to the coin state.** This process can be realized by two different methods.

**Method (i)** Let us return to the subfigure shown in Fig. 1. Adjust the level spacings of qudits $k$ and $j_k$ or adjust the frequency of cavity $C_{j_k}$, such that cavity $C_{j_k}$ is coupled to the $|0\rangle \leftrightarrow |j\rangle$ transition of qudit $j_k$ (qudit $k$) with the coupling strength $g_j$ and the detuning $\Delta_j = \omega_{0j} - \omega_{c_j}$, as shown in Fig. 2. Here, $\omega_{0j}$ is the $|0\rangle \leftrightarrow |j\rangle$ transition frequency of qudit $j_k$ ($k$) while $\omega_{c_j}$ is the frequency of cavity $C_{j_k}$. The Hamiltonian of the $N_k + 1$ qudits ($k$, $1_k$, $2_k$,...,$N_k$) and the $N_k$ cavities ($C_{1_k}$, $C_{2_k}$,...,$C_{N_k}$) in the interaction picture is given by

$$H_I = \sum_{j=1}^{N_k} (g_j a_{j_k}^+ \cdot |0\rangle_k \langle j| + g_j a_{j_k}^+ \cdot |0\rangle_{j_k} \langle j|) e^{-i\Delta_j t} + \text{H.C.} , \tag{8}$$

where the subscript $k$ ($j_k$) in $|0\rangle_k \langle j|$ ($|0\rangle_{j_k} \langle j|$) represents qudit $k$ ($j_k$), and $a_{j_k}$ is the photon annihilation operator of cavity $C_{j_k}$, which is located between qudit $k$ and qudit $j_k$ (Fig. 1). Here H.C. represents Hermitian conjugate.

When $\Delta_j \gg g_j$ (large detuning condition) and the cavities are initially in the vacuum state, the Hamiltonian $H_I$ becomes [45–47]

$$H_e = H_0 + H_I', \tag{9}$$

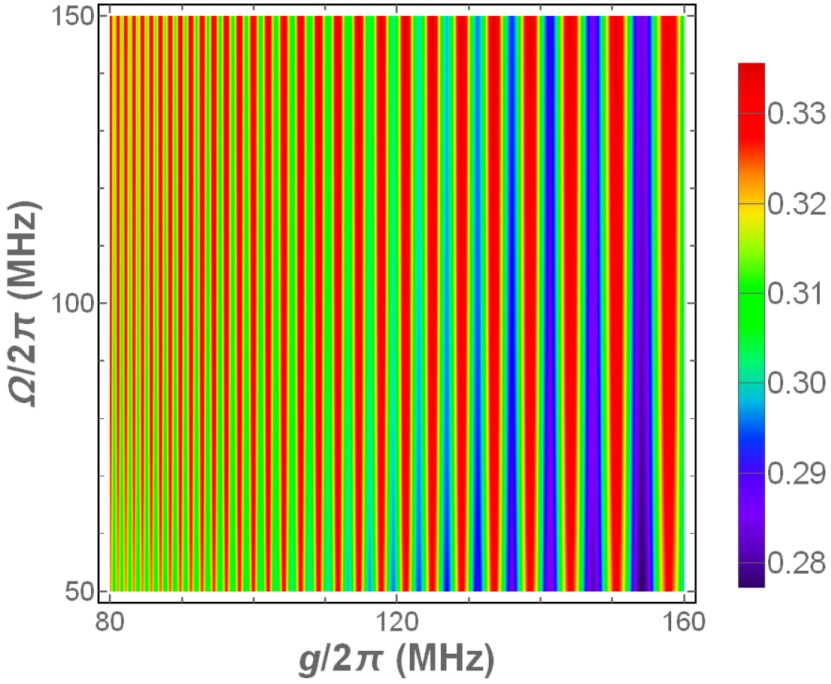

Figure 4: Search probability versus the coupling strength $g$ and the Rabi frequency $\Omega$. Parameters of $\Delta/2\pi = 1$ GHz and $T = 5\,\mu$s are used in the simulation.

with

$$H_0 \;=\; \sum_{j=1}^{N_k}\lambda_j\left(|j\rangle_k\langle j| + |j\rangle_{j_k}\langle j|\right), \tag{10}$$

$$H_I' \;=\; \sum_{j=1}^{N_k}\lambda_j|0\rangle_k\langle j|\cdot|j\rangle_{j_k}\langle 0| + \text{H.C.}\,, \tag{11}$$

where $\lambda_j = g_j^2/\Delta_j$. In a new interaction picture with respect to the Hamiltonian $H_0$, the Hamiltonian becomes

$$\widetilde{H}_e = e^{iH_0 t}H_I'e^{-iH_0 t} = H_I', \tag{12}$$

where we have used the commutation relation $[H_0, H_I'] = 0$.

By setting $\lambda_j = \lambda$ $(j = 1, 2, ..., N_k)$, under the Hamiltonian $\widetilde{H}_e$ and after an evolution time $t = \pi/2\lambda$, the state (7) becomes

$$-\sum_{j=1}^{N_k}\left(\alpha_j|0\rangle_k|j\rangle_{j_k}\prod_{m_k\neq j_k}|0\rangle_{m_k}\right), \tag{13}$$

in which the continued product is taken for $m_k = 1_k, 2_k, ..., N_k$ (without $j_k$). Note that the state (13) has been transformed back to the original interaction picture by performing the unitary transformation $e^{-iH_0 t}$. Equation (13) indicates that the walker, which is initially at position $k$, has moved onto position $j_k$ (occupied by qudit $j_k$ not in the ground state) according to the state $|j\rangle_k$ of the coin.

The main advantages of this method are: The dissipation of the cavities and the crosstalk between the cavities can be ignored since all cavities remain in the vacuum state due to the virtual photon process. We should mention that the condition $\lambda_j = \lambda$ (i.e., $g_j^2/\Delta_j = g_l^2/\Delta_l$, independent of $j$ and $l$) can be met by carefully selecting the detuning $\Delta_j$ ($\Delta_j = \omega_{0j} - \omega_{c_j}$)

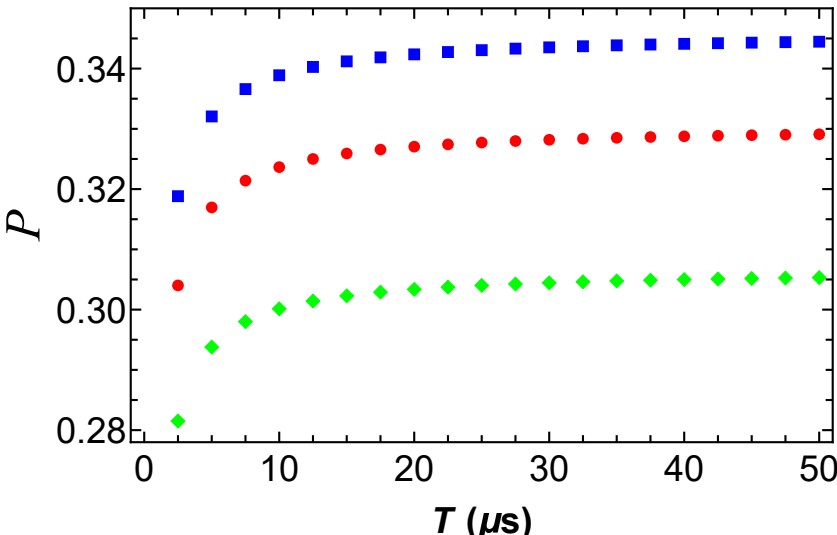

Figure 5: Search probability $P$ versus $T$ for $g/2\pi = 100$ MHz, $\Omega/2\pi = 100$ MHz, and $\Delta/2\pi = 1.0$ GHz. Three lines correspond to the errors 0% (blue square), 5% (red circle) and 10% (green diamond) of the operational time in process II, respectively.

via designing the suitable frequency $\omega_{c_j}$ of cavity $C_{j_k}$ ($j = 1, 2, ..., N_k$). Hence, the method introduced above can be realized in most cavity QED systems.

**Method (ii)** Suppose that cavity $C_{j_k}$ is resonant with the $|0\rangle \leftrightarrow |j\rangle$ transition of qudit $j_k$ and qudit $k$. In this sense, one has $\Delta_j = 0$. The above Hamiltonian $H_I$ in Eq. (8) thus becomes

$$H_I = \sum_{j=1}^{N_k} (g_j a_{j_k}^+ \cdot |0\rangle_k \langle j| + g_j a_{j_k}^+ \cdot |0\rangle_{j_k} \langle j|) + H.C.. \tag{14}$$

With a choice of $g_j = g$, one can easily find that after an evolution time $t = \pi/(\sqrt{2}g)$, the initial state (7) will also evolve to the state (13) [32]. The main advantage of this method is: The operation can be performed at a fast speed due to the use of the qudit-cavity resonant interaction.

So far, we have implemented a one-step DTQW, described by the operator $U_k = S_k \cdot C_k$, on the subgraph shown in Fig. 1. This one-step DTQW is realized through the two processes I and II. In a MD graph, an $M$-step MD DTQW can be realized by repeating the two processes I and II $M$ times in each nodes. By using qudits and cavities, DTQWs in graphs with different structures can be implemented. In general, $n$ qudits and $m$ cavities are required to construct a graph with $n$ nodes and $m$ edges. Moreover, an $(N + 1)$-dimensional qudit is required to encode a node with $N$ neighboring nodes. The operations could become complex when $N$ is large, while large scale regular graphs can be constructed with small $N$ for each nodes. For instance, we show three types of 2D graphs in Fig. 3. Figures 3(a) and 3(b) present two graphs with large scale structures, where each node has four ($N = 4$) and three ($N = 3$) neighbors, respectively. Figure 3(c) gives a compact cubic graph, which can be constructed by a 2D circuit with $N = 3$ [Fig. 3(d)].

## 3 An Application of DTQW on a cubic graph

Circuit quantum electrodynamics (QED), composed of SC qubits and microwave resonators or cavities, has attracted substantial attention because of its controllability, integrability, ready

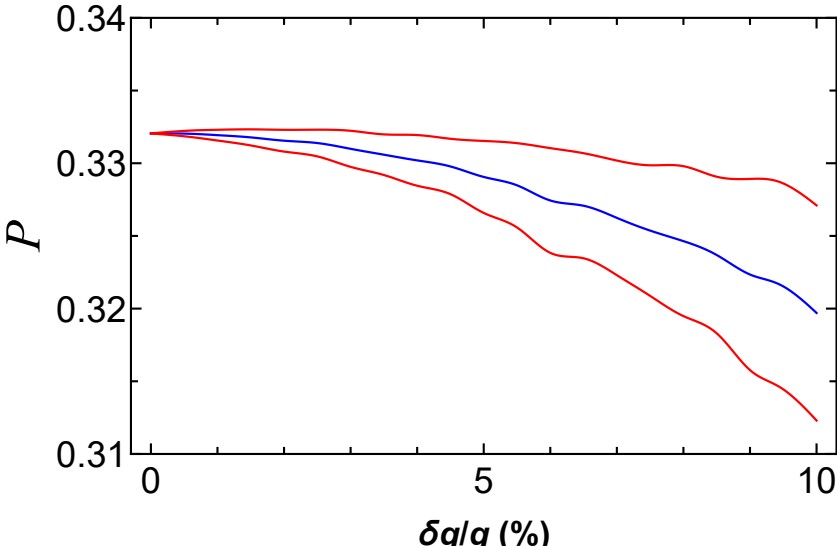

Figure 6: Search probability $P$ versus $\delta g/g$. From up to down, the three lines represent upper bound, lower bound and mean for search probability in 500 random simulations.

fabrication and potential scalability [48–57]. The level spacings of SC devices can be rapidly adjusted (1 ∼ 3 ns) [58–61], and their coherence time has been significantly improved [62–68].

Based on circuit QED, we here implement an eight-element Grover walk search algorithm [14] on a cubic graph [Fig. 3(c)]. A significant advantage of this DTQW-based search algorithm is that it only requires interactions between neighboring qubits (i.e., two-qubit gates), whereas Grover's search algorithm requires multi-qubit controlled gates. Here we numerically simulate this algorithm based on DTQW in a circuit QED system consisting of SC flux qudits and cavities. For details of implementing this search algorithm in circuit QED and its numerical simulation, please refer to appendixes. For a part of numerical calculations, we here use the QuTiP software [43, 44], which is an open-source software for simulating the dynamics of open quantum systems.

The search probability for the target state after two steps, versus the coupling strength $g$ and the Rabi frequency $\Omega$, is plotted in Fig. 4, where the detuning $\Delta/2\pi = 1$ GHz and the decoherence time $T = 5\,\mu s$ are used. Figure 4 indicates that the probability oscillates with $g$ and is not sensitive to $\Omega$. In most of the range of $g$ and $\Omega$ shown in Fig. 4, the probability is close to the ideal probability 0.347. For example, the probability for $g/2\pi = 100$ MHz and $\Omega/2\pi = 100$ MHz is ∼ 0.332.

In Fig. 5, the search probability $P$ versus $T$ is plotted for $g/2\pi = 100$ MHz and $\Omega/2\pi = 100$ MHz. To estimate the effect of imperfect operational time, three lines in Fig. 5 are plotted with a respect to the errors 0%, 5% and 10% of the operational time in process II, respectively. As expected, one can see from Fig. 5 that the probability increases with $T$ (i.e., the decoherence time of the qudits) and is sensitive to the error of the operational time.

To estimate the effect of the possible error in adjusting $g$ ($\Delta$), random errors of $g$ ($\Delta$) in a range $\delta g$ ($\delta\Delta$) are assumed and simulated by 500 times. Lines for upper bound, lower bound and mean of search probability are plotted in Fig. 6 (Fig. 7), by 500 random simulations each for a different error within $\delta g$ ($\delta\Delta$). As $\delta g$ (or $\delta\Delta$) increases, the successful search probability decreases. Moreover, Figs. 6 and 7 show that the effect of the error $\delta\Delta$ is greater than that of the error $\delta g$.

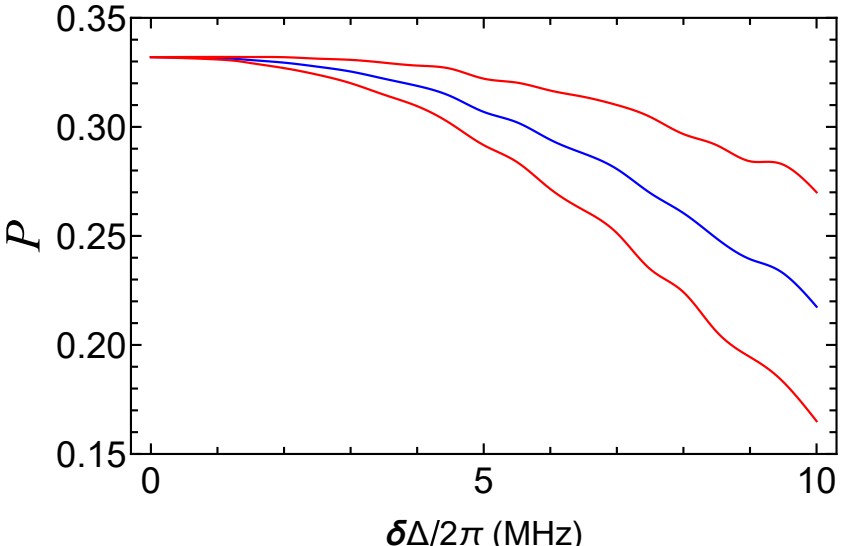

Figure 7: Search probability $P$ versus $\delta\Delta$. From up to down, the three lines represent upper bound, lower bound and mean for search probability in 500 random simulations.

## 4 Conclusion

We have presented a general protocol for implementing DTQWs on multi-dimensional (including 1D) graphs based on cavity QED. As an application of this protocol, we have simulated a Grover walk search algorithm with eight elements in a cubic graph. Numerical simulations show that a high probability (close to the ideal probability) for finding a target element can be achieved with the current circuit QED technology. This protocol is quite general and can be extended to accomplish the same task in various quantum systems, which enable the realization of qudits with multiple energy levels, state transfer between neighboring qudits, and the implementation of the MD coin operator within single qudits.

To our knowledge, our protocol is the first to implement DTQWs on multi-dimensional graphs with different structures using cavity QED. Our protocol can in principle be used to realize multi-dimensional DTQWs, which could have broad applications in quantum computing and quantum simulation. For instance, it could be used for universal quantum computation through two-dimensional DTQWs and for simulating topological phases in two- or three-dimensional quantum systems. Using this protocol, we can further explore the implementation of multi-dimensional DTQWs and investigate the effects of various noises in different cavity QED systems. We anticipate that this research will spark experimental efforts in the near future.

**Funding information**    This work was partly supported by the National Natural Science Foundation of China (NSFC) (Nos. 11074062, 11374083, 11774076, 11974096, U21A20436) and the Innovation Program for Quantum Science and Technology (No. 2021ZD0301705).

## A    Implementation of a Grover walk search

The cubic graph consists of eight SC flux qudits (each with four energy levels $|0\rangle$, $|1\rangle$, $|2\rangle$ and $|3\rangle$) and twelve cavities, as shown in Fig. 3(c). In this graph, each qudit has three neighbor qudits, i.e., the dimension of the coin is three. The coin is encoded through three excited states

$|1\rangle, |2\rangle$ and $|3\rangle$ of a qudit. Suppose all cavities are initially in the vacuum state and qudits are initially prepared in the state

$$|\psi\rangle_0 = \frac{1}{2\sqrt{6}} \sum_{j=1}^{8} \left[ (|1\rangle_j + |2\rangle_j + |3\rangle_j) \prod_{l \neq j} |0\rangle_l \right], \tag{15}$$

where the continued product is taken for $l = 1, 2, ..., 8$ (without $j$). Here and below, the subscript $j$ represents qudit $j$ ($j = 1, 2, ..., 8$). **In this subsection, we will hide the ground states ($|0\rangle$) of all qudits for simplicity**. For example, the initial state (15) will be expressed as

$$|\psi\rangle_0 = \frac{1}{2\sqrt{6}} \sum_{j=1}^{8} \left( |1\rangle_j + |2\rangle_j + |3\rangle_j \right). \tag{16}$$

The initial state (15) of qudits can be prepared from a ground state $|00...0\rangle$ by following steps.

(i) Prepare a state $(|1\rangle_1 + |2\rangle_1 + |3\rangle_1)/\sqrt{3}$ of qudit 1. This can be realized by applying three pulses on qudit 1. First, apply a pulse resonant with the $|0\rangle \leftrightarrow |1\rangle$ transition of qudit 1. The pulse initial phase, the pulse Rabi frequency and the pulse duration are $\{-\pi/2, \Omega, \pi/(2\Omega)\}$. Second, apply a pulse resonant with the $|1\rangle \leftrightarrow |2\rangle$ transition of qudit 1. The pulse initial phase, the pulse Rabi frequency and the pulse duration are $\{-\pi/2, \Omega, \arccos(1/\sqrt{3})/\Omega\}$. Third, apply a pulse resonant with the $|1\rangle \leftrightarrow |2\rangle$ transition of qudit 1. The pulse initial phase, the pulse Rabi frequency and the pulse duration are $\{-\pi/2, \Omega, \pi/(4\Omega)\}$.

In this step, the evolution of the state of qudit 1 is

$$|0\rangle_1 \rightarrow |1\rangle_1 \rightarrow \frac{1}{\sqrt{3}}|1\rangle_1 + \frac{\sqrt{2}}{\sqrt{3}}|2\rangle_1 \rightarrow \frac{1}{\sqrt{3}}(|1\rangle_1 + |2\rangle_1 + |3\rangle_1). \tag{17}$$

(ii) Shift a part of the states $|1\rangle_1$, $|2\rangle_1$, and $|3\rangle_1$ to qudits 2, 4, and 5, respectively, by using Process II introduced in Section 2. According to Eqs. (7) and (13), the state of qudits becomes

$$\frac{1}{2\sqrt{6}}(|1\rangle_1 + |2\rangle_1 + |3\rangle_1) - \frac{\sqrt{3}}{2\sqrt{2}}|1\rangle_2 - \frac{1}{\sqrt{2}}|2\rangle_4 - \frac{1}{\sqrt{2}}|3\rangle_5. \tag{18}$$

(iii) Apply a pulse to each of qudits 2, 4, and 5 to make the transformations $|1\rangle_2 \rightarrow -|2\rangle_2$, $|2\rangle_4 \rightarrow -|3\rangle_4$, and $|3\rangle_5 \rightarrow -|1\rangle_5$. The state of qudits becomes

$$\frac{1}{2\sqrt{6}}(|1\rangle_1 + |2\rangle_1 + |3\rangle_1) + \frac{\sqrt{3}}{2\sqrt{2}}|2\rangle_2 + \frac{1}{\sqrt{2}}|3\rangle_4 + \frac{1}{\sqrt{2}}|1\rangle_5. \tag{19}$$

(iv) Shift a part of the states $|2\rangle_2$, $|3\rangle_4$, and $|1\rangle_5$ to qudits 3, 8, and 6, respectively, by using Process II introduced in Section 2. The state of qudits becomes

$$\frac{1}{2\sqrt{6}}(|1\rangle_1 + |2\rangle_1 + |3\rangle_1) + \frac{1}{2\sqrt{2}}|2\rangle_2 + \frac{1}{2\sqrt{2}}|3\rangle_4 + \frac{1}{2\sqrt{2}}|1\rangle_5 - \frac{1}{2\sqrt{2}}|1\rangle_6 - \frac{1}{2\sqrt{2}}|3\rangle_8 - \frac{1}{2}|2\rangle_3. \tag{20}$$

(v) Apply a pulse to qudit 3 to make the transformations $|2\rangle_3 \rightarrow -|3\rangle_3$. Shift a part of the state $-|3\rangle_3$ to qudit 7 by using Process II introduced in Section 2. The state of qudits becomes

$$\frac{1}{2\sqrt{6}}(|1\rangle_1 + |2\rangle_1 + |3\rangle_1) + \frac{1}{2\sqrt{2}}(|2\rangle_2 + |3\rangle_3 + |3\rangle_4 + |1\rangle_5 - |1\rangle_6 - |3\rangle_7 - |3\rangle_8). \tag{21}$$

(vi) Apply two pulses to qudit $j$ ($j = 2, 3, 4, 5, 6, 7, 8$) to transfer its state to $(|1\rangle_j + |2\rangle_j + |3\rangle_j)/2\sqrt{6}$. Now, the initial state (15) is generated.

Next, using the protocol introduced in Section 2, this search algorithm can be realized by the following two DTQW processes.

**Process I:** Implementing a coin operator

$$C = G \otimes I_n - (G + I_3) \otimes |\tau\rangle_w \langle\tau|, \tag{22}$$

where $I_n$ (with $n = 8$) is the identity matrix in the walker's space, $I_3$ is the identity matrix in the 3D coin space (i.e., the subspace formed by energy levels $|1\rangle, |2\rangle$ and $|3\rangle$ of the qudits), $|\tau\rangle_w$ represents the marked qudit (the element to be searched for), and $G$ is Grover's "diffusion" operator in the coin space

$$G = \begin{pmatrix} -\frac{1}{3} & \frac{2}{3} & \frac{2}{3} \\ \frac{2}{3} & -\frac{1}{3} & \frac{2}{3} \\ \frac{2}{3} & \frac{2}{3} & -\frac{1}{3} \end{pmatrix}. \tag{23}$$

From Eq. (22), one can see that the operator $C$ can be realized by applying $G$ to the subspace of the unmarked qudits and applying a minus identity operator $-I_3$ to the subspace of the marked qudit. Without loss of generality, we assume that the marked element corresponds to qudit 1, i.e., setting $|\tau\rangle_w = |1\rangle_w$.

The operator $G$ can be decomposed as

$$G = \begin{pmatrix} \cos\alpha & -\sin\alpha & 0 \\ \sin\alpha & \cos\alpha & 0 \\ 0 & 0 & 1 \end{pmatrix} \cdot \begin{pmatrix} 1 & 0 & 0 \\ 0 & \cos\beta & -\sin\beta \\ 0 & \sin\beta & \cos\beta \end{pmatrix} \cdot \begin{pmatrix} \cos\gamma & -\sin\gamma & 0 \\ \sin\gamma & \cos\gamma & 0 \\ 0 & 0 & 1 \end{pmatrix}, \tag{24}$$

where $\alpha = 3\pi/4$, $\beta = \arccos(-1/3)$, and $\gamma = \pi/4$. So, the operation of $G$ can be realized by applying three suitable classical pulses.

The operator $-I_3$ can be achieved by applying four pulses to the marked qudit 1. Suppose qudit 1 is in the state $\sum_{j=1}^{3} \alpha_j |j\rangle_1$. Apply a pulse (with an arbitrary initial phase $\phi_1$) to qudit 1. The pulse is resonant with the $|1\rangle \leftrightarrow |4\rangle$ transition of qudit 1, where $|4\rangle$ is an auxiliary energy level of qudit 1. After a pulse duration $t = \pi/(2\Omega)$, the state $\sum_{j=1}^{3} \alpha_j |j\rangle_1$ becomes

$$\alpha_2 |2\rangle_1 + \alpha_3 |3\rangle_1 - i e^{-i\phi_1} \alpha_1 |4\rangle_1. \tag{25}$$

Then, apply a second pulse (with the initial phase $\phi_1$) to qudit 1. The pulse is resonant with the $|1\rangle \leftrightarrow |4\rangle$ transition of qudit 1. After a pulse duration $t = \pi/(2\Omega)$, the state (25) becomes

$$-\alpha_1 |1\rangle_1 + \alpha_2 |2\rangle_1 + \alpha_3 |3\rangle_1. \tag{26}$$

Similarly, apply two pulses to qudit 1 successively. Each pulse is resonant with the $|2\rangle \leftrightarrow |3\rangle$ transition of qudit 1, has an initial phase $\phi_1$, and a duration $t = \pi/(2\Omega)$. After the first pulse, the state (26) becomes

$$-\alpha_1 |1\rangle_1 - i e^{i\phi_1} \alpha_3 |2\rangle_1 - i e^{-i\phi_1} \alpha_2 |3\rangle_1. \tag{27}$$

After the second pulse, the state becomes

$$-\alpha_1 |1\rangle_1 - \alpha_2 |2\rangle_1 - \alpha_3 |3\rangle_1, \tag{28}$$

i.e., the operator $-I_3$ on the marked qudit 1 is realized. Note that a pulse resonant with the $|1\rangle \leftrightarrow |4\rangle$ transition and a pulse resonant with the $|2\rangle \leftrightarrow |3\rangle$ transition can be simultaneously applied onto qudit 1.

In short, the tossing operation of the coin in process I, described by the operator $C$ of Eq. (22), can be implemented by applying three classical pulses to all unmarked qudits and four pulses to the marked qudit.

**Process II:** Implementing the shift operator of the walker [see Fig. 3(c)]

$$
\begin{aligned}
S = & (|2\rangle_w\langle 1| + |4\rangle_w\langle 3| + |6\rangle_w\langle 5| + |8\rangle_w\langle 7|) \cdot |1\rangle_c\langle 1| \\
& + (|4\rangle_w\langle 1| + |3\rangle_w\langle 2| + |8\rangle_w\langle 5| + |7\rangle_w\langle 6|) \cdot |2\rangle_c\langle 2| \\
& + (|5\rangle_w\langle 1| + |6\rangle_w\langle 2| + |7\rangle_w\langle 3| + |8\rangle_w\langle 4|) \cdot |3\rangle_c\langle 3| + H.C..
\end{aligned}
\tag{29}
$$

Here the cavities can be divided into three types according to their frequencies, which are denoted by different color lines [Fig. 3(c)]. This operator $S$ can be realized by applying one of the two methods introduced in Section II. For example, if one adopts the Method (i) introduced above, each term $|i\rangle_w\langle j| \cdot |k\rangle_c\langle k|$ ($k = 1, 2, 3$) and its hermitian conjugate term in Eq. (29) can be realized by the interaction between qudit $Q_i$ and qudit $Q_j$ via the $k^{th}$ type of cavity, with frequency $\omega_{ck} = \omega_{0k} + \Delta_k$. The corresponding Hamiltonian is

$$
H_{ij} = (g_k a_{ij}^+ \cdot |0\rangle_i\langle i| + g_k a_{ij}^+ \cdot |0\rangle_j\langle j|)e^{-i\Delta_k t} + H.C. ,
\tag{30}
$$

where $a_{ij}$ is the photon annihilation operator of the cavity between qudit $Q_i$ and qudit $Q_j$ and $g_k$ is the coupling strength between the cavity and qudit $Q_i$ ($Q_j$). Under the large detuning condition, the Hamiltonian (30) reduces to the following effective Hamiltonian [45–47]

$$
\widetilde{H}_{ij,e} = \lambda_k |0\rangle_i\langle j| \cdot |j\rangle_j\langle 0| + H.C. ,
\tag{31}
$$

where $\lambda_k = g_k^2/\Delta_k$. Under this Hamiltonian, the moving of the walker from position $i$ ($j$) to position $j$ ($i$) is realized after an operational time $t = \pi/2\lambda_k$, only if the coin is in the state $|k\rangle_c$.

The full Hamiltonian corresponding to $S$ is

$$
H_S = \sum_{ij} H_{ij},
\tag{32}
$$

in which the summation is over all terms of $S$ in Eq. (29). By setting all $\lambda_k = g_k^2/\Delta_k = \lambda$ (i.e., $\lambda_1 = \lambda_3 = \lambda_3 = \lambda$), the shift operator $S$ is achieved after an operational time $t = \pi/2\lambda$.

By repeating process I and process II twice, the walker can be searched at the marked position 1 with a relatively high probability. Our calculations show that the ideal search probability is $\sim 0.347$ by applying the operators $C$ and $S$ given above. The search probability of the marked element can be increased by repeating the algorithm [14]. In general, the probability increases with the number of repeating times.

## B   Numerical simulation of the Grover walk search

By considering the finite qudit relaxation, dephasing and photon lifetime, we numerically simulate the Grover walk search algorithm by applying a two-step DTQW. Our simulation is performed by numerically solving the master equation. We adopt SC flux qudits with four energy levels. The energy relaxation time and the dephasing time are assumed as $T_1 = T$ and $T_2 = T/2$. In most simulations, we set $T = 5\ \mu$s, which is a rather conservative consideration since decoherence times ranging from 70 $\mu$s to 1 ms have been reported for a SC flux qudit [69, 70].

The two processes in each step are simulated by the master equation

$$
\begin{aligned}
\frac{d\rho}{dt} = & -i[H, \rho] + \sum_{j=1}^{12} \kappa_{a_j} \mathcal{L}[a_j] \\
& + \sum_{k=1}^{8} \sum_{l=1}^{3} \sum_{m=1}^{l} \gamma_{lm,k} \mathcal{L}[|m\rangle_k\langle l|]
\end{aligned}
\tag{33}
$$

where $\mathcal{L}[\Lambda] = \Lambda\rho\Lambda^+ - \Lambda^+\Lambda\rho/2 - \rho\Lambda^+\Lambda/2$ (with $\Lambda = a_j, |m\rangle_k\langle l|$); $\kappa_{a_j}$ is the decay rate of cavity $j$, which is chosen as $\kappa_{a_j}^{-1} = 1\,\mu s$; $\gamma_{lm,k}$ ($m < l$) is the energy relaxation rate for the level $|l\rangle$ associated with the decay path $|l\rangle \to |m\rangle$ of qudit $k$, which is set as $\gamma_{ml,k}^{-1} = T_1$; and $\gamma_{ll,k}$ (i.e., $m = l$) is the dephasing rate of the level $|l\rangle$ of qudit $k$, which is set as $\gamma_{ll,k}^{-1} = T_2$. In process I, the Hamiltonian $H$ in the master equation is the Hamiltonian describing the interaction between classical pulses and qudits in process I. While in process II, the Hamiltonian $H$ is the $H_S$ given in Eq. (32). Each process is simulated by the master equation, and the density matrix obtained from the previous process is used as the initial density matrix of the next process. For simplicity, in the simulations, we set all coupling strengths between cavities and qudits as $g$, set all detunings as $\Delta$, and set all Rabi frequencies as $\Omega$.

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
