# Peer review of "Implementing discrete-time quantum walks on multi-dimensional arbitrary graphs in cavity quantum electrodynamics"

_SciPost Physics_

## Round 1 · Referee Report · Anonymous (Referee 2) · 2023-12-10

Report

The Authors described a protocol for implementing discrete-time quantum walks (DTQWs) on multi-dimensional (including 1D) graphs. The implementation is described for circuit-QED systems with superconducting qudits and cavities, such that the qudits correspond to the nodes of a given graph, and the cavities serve as the edges connecting these nodes. To show the usefulness of the protocol for implementing quantum algorithms, the Authors conducted simulations of a Grover search algorithm on a cubic graph with eight elements. Their numerical results based on 500 random simulations suggest that the target element can be found with a probability approaching 1 even by assuming experimentally feasible values of relevant parameters. These include: the coupling strength (of $g/2\pi=100$MHz) between the neighboring qudits, the Rabi frequency (of $\Omega/2\pi=100$MHz), detuning (of $\Delta/2\pi=100$GHz), relaxation time $T_1=5\mu s$, and dephasing time $T_2=T_1/2$, which are feasible using the current circuit QED technologies (but seemingly using also other quantum technologies).

(1) I think the theory of quantum walks itself is applied properly. However, possible problems may be in the implementation of QED itself on many qudits. This is crucial, because the manuscript reports (as its main result) a proposal of a circuit-QED implementation, rather than a new theoretical fundamental result.

The analysis of the applied gate operations is quite general, as given in the subsections on Process I (bottom of page 4) and Process II (bottom of page 5). Thus, it seems that the operations are not limited to circuit-QED implementations. In my opinion, similar values of the above-mentioned parameters are also experimentally feasible for trapped ions or other systems allowing to experimentally reach the ratio of $g/\Omega=1$, as assumed in Fig. 5. Anyway, I would suggest to clarify the issue whether it is possible or not (see also the comments below) to use other platforms by applying the transformations described in the above-mentioned subsections.

(2) The assumed qudits must have many levels (more than 3) and the operations require many sequences and a very high precision. I do not know to what extent this is currently achievable, but the Authors should clearly address this issue in the manuscript.

(3) The Authors mentioned that their method allows for walks on arbitrary graphs, but they only discussed the construction of a single element from which the whole graph can be assembled. It is like discussing only two qubits together with 1- and 2-qubit gates, and then saying that a quantum computer can be assembled from those. Although this is mathematically correct, but extremely challenging concerning any physical implementations. Considering the complexity of single operations (see the previous point), the whole system will be even more prone to imperfections.

(4) Although the Authors consider a simulation for Grover's search walk and consider simple noise models, this analysis is done too superficially in my opinion.

(5) I feel that the paper in its present form does not discuss a realistic circuit-QED implementation. The main result is effectively a decomposition of abstract unitarity operations, which are needed to realize a quantum walk, into sequences that can be theoretically realized in various platforms.

The discussion of realistic constraints and noise effects is very limited. Indeed, its considered via a general Lindblad master equation (in Appendix B), which can be applied in the same form for implementations using other platforms and just by modifying the values of the relevant parameters. Thus, a Reader of the manuscript would like to know some more details specific to the chosen circuit-QED platform.

Finally, I must admit that the paper is relatively clearly and consistently written, so one can easily follow the presentation and understand the protocol even concerning many details.

In conclusion, I could recommend the publication of this work in SciPost if the manuscript was adequately revised according to at least some of the above-mentioned issues.

---

## Round 2 · Referee Report · Anonymous (Referee 2) · 2024-11-7

Report

The first-round reports from both referees were positive and supportive of the article’s publication, provided that specific revisions were made. I have carefully reviewed both the original and revised versions of the manuscript, paying particular attention to the referees' initial feedback. I believe that the revised paper effectively addresses their critical comments and has been significantly improved to meet SciPost's high publication standards. I therefore recommend the revised article for publication in its current form in SciPost Physics.

Recommendation

Publish (easily meets expectations and criteria for this Journal; among top 50%)

---

## Round 2 · Referee Report · Anonymous (Referee 1) · 2024-11-11

Strengths

Story nicely told with good illustrations.

Weaknesses

1- Unclear novelty. For example, with a quick search I stumbled across
[A] P. P. Rohde, A. Schreiber, M. Štefaňák, I. Jex, and C. Silberhorn, "Multi-walker discrete time quantum walks on arbitrary graphs, their properties and their photonic implementation", New J. Phys. 13 (2011) 013001
[B] L. Razzoli, G. Cenedese, M. Bondani, and G. Benenti, "Efficient Implementation of Discrete-Time Quantum Walks on Quantum Computers", Entropy 26 (2024) 313
I am neither insisting that these are indeed so relevant that they need to be cited, nor that the list is complete. However, if it is easy to find references that appear to be relevant (and are more than 10 years old), it is not convincing that this is as uncharted territory as the authors pretend.

2- Relevance unclear. The proposal in the present work requires multi-level systems and many cavities to couple them. A reliable realization of these will be challenging and I have seen no evidence that multi-dimensional quantum walks can indeed be expected to perform better than more standard uses of the corresponding platforms.
In this respect, the beginning of the second paragraph on page 9 is telling. The authors do not "implement" the circuit experimentally, but only simulate it. Furthermore an eight-element search (three qubits in the conventional Grover search algorithm) is not a relevant problem size.

Report

This work makes a theoretical suggestion of a possible implementation of discrete-time quantum walks on multi-dimensional graphs in cavity quantum electrodynamics. I have some reservations concerning the relevance of this submission, see "Weaknesses". However, the work is what it is and it is well presented. I thus suggest that the appropriate venue would be SciPost Physics Core rather than SciPost Physics.

I have a number of further comments of a more typographic nature that I list as "Requested changes".

Requested changes

1- Double-check completeness of the list of references.
2- The abbreviation "MD" appears in the Table of Contents on page 1 before it is introduced.
3- Why do the authors use subscripts "$w$" and "$c$" in Eqs. (1), (2), etc. only for kets $\vert \cdot \rangle$ and not for bras $\langle\cdot\vert$?
4- Figure placement should be revisited. E.g., Fig. 2 is placed on page 5, but not discussed before page 6, Fig. 3 is placed on page 6, but not discussed before page 8, Fig. 4 is placed on page 7, but not discussed before page 9, etc.
5- Eq. (8): I suspect that the meaning of the subscripts of the kets $\vert \cdot \rangle$ has changed; in the appendices (e.g., Eq. (15)) yet further notations may be used, in addition to the original one (see, e.g., Eq. (22)). Please double-check and make sure to avoid ambiguous notation.
6- Unify typesetting of "H.C.": "${\rm H.C.}$" in Eqs. (8) (where it is actually defined) and (11) versus "$H.C.$" in Eqs. (14), (29), (30), and (31).
7- First line of caption of Fig. 5: explain meaning of "$T$".
8- Related to the previous point: In the fourth paragraph on page 9, "$T$" is used before it as explained as "the decoherence time of the qudits".
9- First lines of captions of Figs. 6 and 7: "up" $\to$ "top", "down" $\to$ " bottom.
10- Line below Eq. (16): Insert article "the" between "by" and "following".
11- Why not typeset Eq. (33) on one line?
12- I believe that Refs. [43,44] are cited only after Ref. [69] and should thus be moved there.
13- "Including the DOI is particularly important: the paper can only be published if references are externally linked", see https://scipost.org/SciPostPhys/authoring#manuprep. $\Rightarrow$ Please add DOIs to all references whenever possible.
14- There are other typographic issues such as a comma "," before an "and" in the list of three items at the end of the first paragraph of the Conclusions (American English) while at other places there is no such comma (British English).

Recommendation

Accept in alternative Journal (see Report)

---

## Round 2 · Author Response

Dear editor,

Thank you for providing us with feedback from the reviewers. We have given a considerable revision on the manuscript according to their comments and suggestions. The following are our replies to their comments in order.

Our reply to Reviewer 1:

We are pleased to see your positive comments on our work and thank you for recommending publication of our paper after the issues are addressed or clarified.

(1) You are correct that our theoretical protocol is general and can be implemented in other quantum systems, such as trapped ions. In the previous manuscript version, we implemented this protocol in circuit QED primarily due to our familiarity with this system.

In Section 2 of the revised manuscript, we first present a theory for the implementation of multi-dimensional DTQWs based on cavity QED, which can be applied in various platforms including circuit QED and trapped ions. Then, we numerically simulate a search process in circuit QED based on our protocol. We have made corresponding adjustments throughout the entire manuscript.

(2) We have appended the following content to the end of paragraph 4 in the Introduction:“the operations in qudits (i.e., quantum systems with more than two energy levels) and their applications in quantum algorithms have attracted considerable interests in recent years, with experimental realizations achieved in various systems [34-40]. Compared to a two-dimensional qubit, a qudit offers a higher-dimensional space for encoding more information.”. Additionally, we have incorporated seven related references [34-40] in the revised manuscript.

(3) We agree with you that our protocol is still in its preliminary theoretical stage and requires further research before it can be applied to graphs in experiments. We have removed the word "arbitrary" from the phrase "arbitrary graphs" throughout the manuscript.

(4) Since we are mainly familiar with cavity QED theories and don't have a corresponding lab, it's difficult for us to conduct in-depth simulations related to the experimental implementation of Grover's search walk. We apologize for the limitations of our simulations.

(5) We have revised the manuscript to introduce a theoretical protocol for implementing DTQW in multi-dimensional graphs using cavity QED. Due to time constraints and our lack of familiarity with other platforms, our discussion in this manuscript is limited. However, we plan to explore realistic constraints and noise effects on various platforms in our future works.

Our reply to Reviewer 2:

We appreciate your positive feedback on our work and have carefully revised the manuscript based on your suggestions.

(1) We have included an introduction detailing the preparation of the initial state (15), located below Equation (15) in Appendix A. If the initial state in a quantum information task is a superposition state of qudits like Equation (15), its preparation is similar to the process outlined in Appendix A. However, if the initial state is an entangled state, its preparation will differ depending on the specific scenario.

(2) We have incorporated the suggested content in the final paragraph of Section 2 of the revised manuscript.

(3) We have included the suggested content in the last paragraph of Section 4 of the revised manuscript.

(4) We have added two references [43] and [44], and inserted a sentence in the second paragraph of Section 3 stating “For a part of numerical calculations, we here use the QuTiP software [43,44], which is an open-source software for simulating the dynamics of open quantum systems.”

(5) We have added 15 recent and relevant references [2-7,25,30,34-40] to the revised manuscript.

(6) We have incorporated the suggested content in the final paragraph of Section 4 of the revised manuscript.

---

## Round 2 · List of Changes

(1) We have replaced “circuit quantum electrodynamics” with “cavity quantum electrodynamics” in the title, and made corresponding adjustments throughout the entire manuscript.
(2) We have removed the word "arbitrary" from the phrase "arbitrary graphs" throughout the manuscript.
(3) We have added a new paragraph introducing cavity QED at the beginning of the Introduction and have cited the relevant references.
(4) We have appended the following content to the end of paragraph 4 in the Introduction: “the operations in qudits (i.e., quantum systems with more than two energy levels) and their applications in quantum algorithms have attracted considerable interest in recent years, with experimental realizations achieved in various systems [34-40]. Compared to a two-dimensional qubit, a qudit offers a higher-dimensional space for encoding more information.”. Additionally, we have incorporated seven related references [34-40] in the revised manuscript.
(5) We have incorporated the content in the final paragraph of Section 2 to present the resource requirements and the complexity of the required circuitry.
(6) We have included the content in the last paragraph of Section 4 to present the potential applications of the protocol and outline its future research directions.
(7) We have added two references [43] and [44], and inserted a sentence in the second paragraph of Section 3 stating “For a part of numerical calculations, we here use the QuTiP software [43,44], which is an open-source software for simulating the dynamics of open quantum systems.”
(8) We have added 15 recent and relevant references [2-7,25,30,34-40].
(9) We have included an introduction detailing the preparation of the initial state (15), located below Equation (15) in Appendix A.

---

## Editorial Decision

awaiting_resubmission